# An Innovative Inhibitor with a New Chemical Moiety Aimed at Biliverdin IXβ Reductase for Thrombocytopenia and Resilient against Cellular Degradation

**DOI:** 10.3390/pharmaceutics16091148

**Published:** 2024-08-30

**Authors:** Hoe-Myung Jung, Jung-Hye Ha, Mark Vincent C. dela Cerna, Joseph A. Burlison, Joonhyeok Choi, Bo-Ram Kim, Jeong Kyu Bang, Kyoung-Seok Ryu, Donghan Lee

**Affiliations:** 1Korea Basic Science Institute, 162 Yeongudanji-Ro, Ochang-Eup, Cheongju-Si 28119, Republic of Korea; jhm2022@kbsi.re.kr (H.-M.J.); jhyeokchoi@kbsi.re.kr (J.C.); brk9126@kbsi.re.kr (B.-R.K.); bangjk@kbsi.re.kr (J.K.B.); 2Department of Bio-Analytical Science, University of Science & Technology, Daejoen 34113, Republic of Korea; 3New Drug Development Center, Daegu-Gyeongbuk Medical Innovation Foundation (DGMIF), 80 Cheombok-ro, Dong-gu, Daegu 41061, Republic of Korea; hjh0227@kmedihub.re.kr; 4Department of Biochemistry, Chemistry and Physics, Georgia Southern University, 11935 Abercorn Street, Savannah, GA 31419, USA; mdelacerna@georgiasouthern.edu; 5Department of Medicine, James Graham Brown Cancer Center, University of Louisville, 505 S. Hancock St., Louisville, KY 40202, USA; joe.burlison@louisville.edu; 6Dandicure Inc., Ochang-Eup, Cheongju-Si 28119, Republic of Korea

**Keywords:** biliverdin IXβ reductase-inhibitor complex structure, azoreductase-resistant inhibitor, thrombocytopenia, NMR spectroscopy, X-ray crystallography

## Abstract

Biliverdin IXβ reductase (BLVRB) has emerged as a promising therapeutic target for thrombocytopenia due to its involvement in reactive oxygen species (ROS) mechanisms. During the pursuit of inhibitors targeting BLVRB, olsalazine (OSA) became apparent as one of the most potent candidates. However, the direct application of OSA as a BLVRB inhibitor faces challenges, as it is prone to degradation into 5-aminosalicylic acid through cleavage of the diazenyl bond by abundant azoreductase (AzoR) enzymes in gut microbiota and eukaryotic cells. To overcome this obstacle, we devised olsalkene (OSK), an inhibitor where the diazenyl bond in OSA has been substituted with an alkene bond. OSK not only matches the efficacy of OSA but also demonstrates improved stability against degradation by AzoR, presenting a promising solution to this limitation. Furthermore, we have found that both OSK and OSA inhibit BLVRB, regardless of the presence of nicotinamide adenine dinucleotide phosphate, unlike other known inhibitors. This discovery opens new avenues for investigating the roles of BLVRB in blood disorders, including thrombocytopenia.

## 1. Introduction

Platelets, also known as thrombocytes, originate from megakaryocytes (MK), which differentiate from multipotent hematopoietic stem cells [1,2]. In addition to their roles in blood hemostasis and pathological thrombosis, platelets are also involved in inflammation, neoangiogenesis, innate immunity, adaptive immune response, and tumor metastasis [3,4,5]. Consequently, modulating platelet responses and regulating their population represent significant therapeutic strategies against platelet disorders [6]. Dysregulation of blood platelets may cause bleeding disorders [7,8]. One such disorder is thrombocytopenia, a genetic bleeding disorder caused by reduced platelet counts [9]. Typical interventions for thrombocytopenia involve increasing platelet counts through platelet generation [10], induction, or transfusion [11]. Conversely, thrombocytosis involves excessive platelets in the blood, contrasting with thrombocytopenia symptoms [12]. A large-scale transcriptomic analysis on essential and reactive thrombocytosis cohorts identified biliverdin IXβ reductase (BLVRB) as a regulator of platelet production through MK differentiation by controlling reactive oxygen species (ROS) [13,14].

In the pathway of heme degradation, BLVRB utilizes NAD(P)H downstream of heme oxygenase(s)-1 (inducible HMOX1) and -2 (constitutive HMOX2) to reduce biliverdin (BV)-IXβ to bilirubin (BR)-IXβ. BR, the product of this process, acts as a potent antioxidant [15,16] and exhibits apparent cytoprotective effects [17], although high concentrations can lead to toxicity (hyperbilirubinemia). Consequently, the BV/BR redox cycle, under the control of BLVRB, plays a crucial role in regulating reactive oxygen species (ROS) [18]. Studies involving induced pluripotent stem cells (iPSCs) expressing a loss-of-function mutant of BLVRB (BLVRBS111L) have demonstrated ROS accumulation and significantly increased proliferation, as measured by MK-colony-forming (CFU-MK), in modified CD34+/BLVRBS111L hematopoietic stem cells (HSCs). In contrast, wild-type BLVRB does not exhibit these effects [14]. This loss-of-function mutant (BLVRBS111L) induces MK differentiation and consequently promotes platelet production through ROS accumulation [13,19]. Leveraging the role of BLVRB in this pathway is a novel strategy to increase platelet count though the removal of antioxidant BR by inhibition of BLVRB reductase activity [14].

Several compounds, notably xanthene dyes and acridine-based molecules, which are structurally similar to BLVRB substrate flavin mononucleotide (FMN), have been shown to inhibit BLVRB [20]. Erythrosin B and phloxine B, two of the most potent BLVRB inhibitors, localize in the BLVRB active site [21,22]. Furthermore, structural studies have revealed active site hydrogen bond networks and elucidated the critical role of S111 in catalysis [23,24,25]. Despite binding to the active site, enzyme kinetics studies have shown that these inhibitors act in a noncompetitive manner [22,25,26]. Both erythrosin B and phloxine B have been utilized as food colorants, with phloxine B also exhibiting antimicrobial activity against various Gram-positive bacteria [27]. There are some limitations to the potential use of both inhibitors in the clinical setting. Chronic administration of erythrosin B in rats results in thyroid stimulation, which may promote thyroid tumor growth [28]. Additionally, through meticulous investigation using NMR and dynamic light scattering (DLS) experiments, it has been confirmed that both erythrosin B and phloxine B induce the multimerization of BLVRB [29]. Hence, there is a need for new drug candidates targeting BLVRB to treat platelet disorders.

In pursuit of this goal, a drug repurposing approach has been employed to screen for new candidates, resulting in the identification of 20 potential inhibitors [30]. However, this method may be less attractive to pharmaceutical companies, as generic products with the same active pharmaceutical ingredient (API) can still be used off-label in clinical practice [31]. Thus, there is a desire for novel compounds. Among the inhibitors derived from the repositioning of FDA-approved drugs, olsalazine exhibits the strongest affinity for inhibition [30]. However, reports from recent studies further substantiate this susceptibility by demonstrating olsalazine’s degradation in various human gastrointestinal environments [32,33]. Olsalazine contains a diazenyl bond susceptible to cleavage by abundant Azoreductase (AzoR) [34]. It has also been reported that olsalazine has a plasma elimination half-life of approximately 1 h [35].

Consequently, our focus shifted towards developing a new compound that is not subject to cleavage by AzoR while retaining similar inhibitory, biochemical, and biophysical properties. To this end, we devised a new chemical scheme to replace the diazenyl bond of olsalazine with an alkene bond, resulting in the creation of olsalkene (OSK). Through NMR chemical shift perturbation (CSP) analysis, we elucidated the binding mode of OSK to the active site of BLVRB. Detailed biophysical characteristics of BLVRB-binding were assessed using isothermal titration calorimetry (ITC). Furthermore, we determined the structure of the BLVRB-OSK complex, which corroborates the binding mode identified through NMR studies. These discoveries elucidate the mechanism of action of these potential drug candidates and provide novel modalities that can be tested in vitro and in preclinical studies.

## 2. Materials and Methods

### 2.1. Chemicals

All chemicals, including olsalazine, phloxine B, FMN, NADH, NAD^+^, NADPH, NADP^+^, and methylhydroquinone (2-MHQ), were purchased from Sigma Aldrich (St. Louis, MO, USA), unless otherwise specified. For synthesis of olsalkene, high-purity (>99.9%) organic solvents, reagents, and starting materials from Sigma-Aldrich, TCI were used as is. ^15^N-ammonium chloride, ^13^C-glucose, and ^2^H-dimethylsulfoxide (d_6_-DMSO) were purchased from Cambridge Isotope Laboratories. All compounds in the manuscript are >95% pure by NMR analysis.

### 2.2. Chemical Synthesis of Olsalkene (OSK)

Merck aluminum sheets with silica gel 60 F_254_ were used for thin-layer chromatographic analyses. Visualization was performed by UV light and staining with phosphomolybdic acid and potassium permanganate. Merck silica gel 60 (230–400 mesh) was used for compound purification by column chromatography. All proton NMR spectra were recorded using a Bruker 400 MHz NMR. Chemical shifts are reported in parts per million (ppm) relative to an internal standard. Product mass was determined by Shimadzu (Kyoto, Japan) (MALDI-TOF) mass spectrometer. 

We have synthesized the olsalkene using the following method (Figure 1). To a stirred solution of 5-formyl-2-hydroxybenzoic acid (1 g, 0.0060 mole), methyl iodide (1.49 mL, 0.0240 mole) and dried K_2_CO_3_ (2.91 g, 0.0210 mole) were added in dry DMF (10 mL) solvent contained in a 100 mL two-necked round-bottomed flask kept under nitrogen atmosphere at overnight in a magnetic stirrer; the progress of the reaction was monitored through TLC analysis. After completion, the crude mass was quenched with water (3 × 25 mL) and extracted with dichloromethane, and purification on column chromatographic method yielded methyl 5-formyl-2-methoxybenzoate (950 mg, 95%). Apart, Zn powder (1.01 g, 0.0224 mole) was taken in THF at 0 °C, and TiCl_4_ (2.12 g, 0.0112 mole) was added for 30 min successively. The solution was warmed from room temperature to 60 °C for 2 h, the reaction mixture was again cooled, and methyl 5-formyl-2-methoxybenzoate (500 mg, 0.0028 mole) in THF was added dropwise, which refluxed for 4 h, was further quenched with K_2_CO_3_, and extracted with DCM, providing (E)-dimethyl 5,5′-(ethene-1,2-diyl)bis(2-methoxybenzoate) (150 mg, 15%). To a solution of (E)-dimethyl 5,5′-(ethene-1,2-diyl)bis(2-methoxybenzoate) (100 mg, 0.0003 mole) in DCM at −78 °C, BBr_3_ (240 µL, 0.0014) was added in a 10 mL round-bottomed flask and allowed to stir in room temperature for 5 h. Then, 1 mL of 1 M HCl was added for quenching and was stirred well for 15 min. Then, the solvent was removed under reduced pressure, and the crude mass was dissolved in ethyl acetate and washed twice with water and brine solution. The resulting mixture was dissolved in 95% ethanol, followed by potassium hydroxide solution addition, wherein it was allowed to stir for 3 h under reflux condition. After precipitation occurred, the reaction mass was again acidified with 1 M HCl and brine solution, extracted, and underwent recrystallization from ethyl acetate solvent. Finally, the colorless crystalline solid of the (E)-5,5′-(ethene-1,2-diyl)bis(2-hydroxybenzoic acid) (60 mg, 72%) was achieved and further confirmed by ^1^H NMR and mass spectra analyses (purity > 99%). ^1^H NMR (CDCl_3_, 400 MHz) δ 6.96 (d, 2H, *J* = 8.64 Hz), 7.12 (s, 2H), 7.79 (d, 2H, *J* = 8.56 Hz), 7.96 (s, 2H); MALDI-TOF *m*/*z* calcd for C_16_H_12_O_6_: 300.27, found 322.5044 (M+Na), respectively.

### 2.3. Expression and Partial Purification of AzoR

*E. coli* DH5α cells were cultured in LB medium to obtain AzoR protein at 37 °C. Since various electrophilic quinones increase the mRNA level of the acpD gene-encoding AzoR protein [36], 0.5 mM methylhydroquinone (2-MHQ) was added to the culture when the optical density of the cells at 600 nm (OD_600_) reached 0.8. The cells were harvested by centrifugation after ~3–4 h. To enrich the AzoR protein, anion-exchange column chromatography using the Hitrap-Q HP column (Cytiva, Marlborough, MA, USA) was performed, in which the column was pre-equilibrated with buffer (pH 8.0, 25 mM Tris-HCl), and then 1.0 M NaCl gradient was applied to elute proteins. The AzoR activity in the elution fractions was measured by the enzyme activity assay using 0.1 mM menadione and 0.1 mM NADH, in which decreasing absorption at 340 nm was measured following the previously reported method [37]. A 0.1 M stock solution of menadione was prepared in 100% DMSO. The elution fractions were diluted 100-fold before the assay, in which 1% DMSO was additionally added to dissolve the 0.1 mM menadione. Finally, the elution fraction of the highest activity was used for the cleavage assay of olsalazine after 1000-fold dilution in phosphate-based saline (PBS) buffer. The fraction containing AzoR was used without further purification because the sole purpose of AzoR is to check the cleavage of the diazenyl bond in olsalazine.

### 2.4. Expression and Purification of BLVRB

The human BLVRB protein was prepared following the previously reported method [30]. Briefly, the N-terminal His6-tagged BLVRB was expressed in the *E. coli* BL21(DE3) strain using the pET-21b protein expression vector. The transformed *E. coli* cells were grown in LB and M9 media at 37 °C to express the nonlabeled and isotope-labeled (^15^N or ^13^C/^15^N) BLVRB proteins, respectively. To induce protein expression, 0.5 mM IPTG was added to the cultures when the OD_600_ reached 0.8, and then the cells were harvested after ~3–4 h. Cells were lysed by sonication in buffer (pH 8.0, 25 mM Tris-HCl, 500 mM NaCl, and 10 mM β-mercaptoethanol). The BLVRB protein was first purified by His-tag affinity column chromatography using the HisTrap HP column (Cytiva). To remove the His-tag, the protein elution was dialyzed in buffer (pH 8.0, 50 mM Tris-HCl, and 1 mM dithiothreitol) overnight at 4 °C, after adding thrombin (~5 units per mg protein). Since thrombin bound to HiTrap-Q HP column at pH 8.0, ion-exchange column chromatography was performed with buffer (pH 8.0, 25 mM Tris-HCl, and 1 mM dithiothreitol), and then the elution was performed with 1.0 M NaCl gradient; BLVRB and thrombin eluted at ~150 and ~300 mM NaCl, respectively. To separate the His_6_-tagged BLVRB from the completely cleaved BLVRB, the HisTrap HP column was tandemly attached to the next of the HiTrap-Q column. The BLVRB protein was further purified by size exclusion chromatography (SEC) using the HiLoad 16/600 Superdex 75 (Cytiva) with nonbuffered solution (50 mM NaCl and 1 mM dithiothreitol). The protein fractions were concentrated using an ultracentrifugal filter with 10 kDa MWCO (Millipore, Burlington, MA, USA) and then were stored at −70 °C before use. Apo-BLVRB was prepared using the previously reported refolding method [38], with additional column chromatography steps. Briefly, the purified BLVRB was denatured in buffer (pH 7.5, 25 mM Tris-HCl, 10 mM β-mercaptoethanol, and 6.0 M urea) and then applied to the HisTrap HP column. After washing with buffer (pH 5.0, 25 mM Na-acetate, 10 mM β-mercaptoethanol, and 6.0 M urea), they were eluted with an additional 1.0 M NaCl gradient. After 2-fold dilution with buffer (pH 5.0, 20 mM Na-acetate, 10 mM β-mercaptoethanol, and 6.0 M urea), apo-BLVRB was further purified by cation-exchange column chromatography using the HiTrap-SP column (Cytiva). Protein elution was performed by 1.0 M NaCl gradient in the presence of 6.0 M urea. The protein was refolded by dialysis in buffer (pH 6.5, 50 mM Bis-tris, 150 mM NaCl, and 1 mM DTT). After removing the aggregate (~20%) by centrifuge, it was purified by SEC using the HiLoad 16/600 Superdex 75 with buffer (pH 6.5, 10 mM Bis-tris, 50 mM NaCl, and 1 mM DTT). The apo-BLVRB solution was concentrated and then stored at −70 °C. Although thawing the frozen sample caused ~10% precipitation of apo-BLVRB, the maintenance of the structure was confirmed by ^1^H-^15^N HSQC spectrum. The concentration was also determined using UV absorption with an extinction coefficient (A_280_) of 14,440 M^−1^⋅cm^−1^. The purity was confirmed by SDS-PAGE gel analysis.

### 2.5. Crystallization and Structure Determination of the BLVRB:OSK Complex

Overall, protocols to crystallize BLVRB protein and soak the crystals with the OSK molecule were very similar to the previously reported method [30]. Briefly, 1 mM NADPH was added to the solution of BLVRB protein and then was further purified by SEC in buffer (pH 8.0, 20 mM Tris-HCl, 150 mM NaCl, and 1 mM DTT) using the HiLoad 16/600 Superdex 75 (Cytiva, Amersham, UK). The protein fractions were concentrated using the ultracentrifugal filter of 10 kDa MWCO. The protein stock solution (14.8 mg/mL) was aliquoted and then stored at −70 °C before usage. The BLVRB crystals were grown in buffer (pH 6.5, 0.1 M Bis-Tris, and 1.9~2.0 M ammonium sulfate) using the hanging-drop vapor-diffusion method at 18 °C. For soaking of the OSK molecule, the crystals were incubated in the same crystallization buffer, including 1 mM OSK and 0.2 mM NADP^+^ for 1 day at 18 °C. A total of 1% DMSO was additionally present in the soaking solution, since the OSK stock solution (100 mM) was prepared in 100% DMSO. The crystals of the BLVRB:OSK complex were looped and flash-frozen in liquid nitrogen, in which 20~25% glycerol was used as a cryo-protectant. The X-ray diffraction data were collected on the beamline 5C at the Pohang accelerator laboratory. The diffraction data were processed using the HKL-2000 program [39], and then the structure was solved by molecular replacement (MR) using the previously reported coordinate of BLVRB (PDB code, 7ERA) [30]. The structure model was improved by alternating cycles of model building using the Coot 0.9.8.92 program [40] with refinement using the PHENIX 1.21.2-5419 software package [41]. The structural visualizations were performed using the Chimera program [42]. Schematic diagrams of the interaction networks about OSK and OSA in complex with BLVRB were generated using LIGPLOT^+^ v.2.2 [43].

### 2.6. Enzyme Activity Assays

Enzyme kinetics were monitored using the absorption of NADPH at 340 nm with the SpectraMax Gemini EM Microplate Reader (Molecular Devices, San Jose, CA, USA). The enzyme reactions were conducted with 0.5 µM BLVRB, 100 µM FMN, and 100 µM NADPH in PBS buffer containing 0.5% DMSO in the presence of various concentrations of inhibitors (OSK and OSA). The initial velocities of the reactions were estimated from data collected over 5 min and were fitted to the Michaelis–Menten equation. The inhibition model and inhibition constants were determined using the in-house Python script of nonlinear least squares regression fitting.

### 2.7. Isothermal Titration Calorimetry (ITC)

The ITC experiment was conducted in buffer (pH 6.5, 50 mM Bis-tris, 50 mM NaCl, and 0.1 mM TCEP) using a Microcal Auto-iTC200 (Malvern Instruments, Malvern, UK) at 25 °C. The calorimetric cell (200 μL) and syringe (40 μL) were loaded with 0.09 mM BLVRB and 1.1 mM drug (OSK or OSA), respectively. The ITC data were analyzed after subtracting the heat from blank injections that were measured under the same conditions but in the absence of BLVRB. Data processing was carried out using the Origin software provided by the manufacturer, where the fittings were performed with a one-site binding model.

### 2.8. NMR Experiments

All NMR experiments were performed with the Bruker 800 MHz spectrometer at 25 °C. A total of 0.2 mM ^15^N-labeled BLVRB or apo-BLVRB was prepared in buffer (pH 6.5, 50 mM Bis-tris, 50 mM NaCl, and 5% D2O), and their ^1^H-^15^N heteronuclear single-quantum coherence (HSQC) spectra were recorded in the presence and absence of drug molecules. The 1-dimensional (1D) ^1^H spectra to monitor the enzyme reactions were conducted in PBS buffer. The chemical shift perturbation (CSP) data were presented using the square root of (ΔH)^2^ + [(ΔN)/6]^2^ to normalize the relative effects from the CSPs of ^1^H and ^15^N [44]. The diffusion-ordered spectroscopy (DOSY) spectra were measured using the ledbpgppr2s, with 80 ms diffusion delay, 2 s relaxation delay, and 32 gradient strength increments. The backbone assignments were referenced from BMRB Entry ID: 27463 for holo BLVRB and Entry ID: 27462 for apo BLVRB to assign the spectra in the presence and absence of the drug.

## 3. Results and Discussion

### 3.1. Enhancing Stability: Substituting Diazenyl Bond with Alkene Bond to Prevent Azoreductase Cleavage

The revelation of biliverdin IXβ reductase (BLVRB) as a promising target for thrombocytopenia treatment has sparked endeavors to create inhibitors. Recently, olsalazine (OSA) has emerged as a top contender through repurposing FDA-approved drugs for this purpose. Since initially prescribed for ulcerative colitis as an anti-inflammatory agent, OSA is metabolized by Azoreductase (AzoR) [34], and the cleavage of OSA to 5-aminosalicylic acid (5-ASA) was tested by using the partially purified Escherichia coli (*E. coli*) AzoR (Appendix A). The expression of AzoR was enhanced by adding methylhydroquinone (2-MHQ) to the cell culture [36]. Indeed, the incubation of OSA with the column fraction containing AzoR activity and NADPH lead to the cleavage of its diazenyl bond and the formation of 5-ASA (Figure 1 and Appendix A), the active compound responsible for its therapeutic effects. Thus, employing OSA as a BLVRB inhibitor in vivo poses uncertainties due to this bond cleavage. Furthermore, the binding of 5-ASA, the products of AzoR catalysis, to BLVRB is reduced significantly (Appendix A). Therefore, it is crucial to create a noncleavable compound. To achieve this, we developed a new synthetic scheme (Appendix A) that replaces the diazenyl bond with an alkene bond while preserving the trans configuration (Appendix A). This yields olsalkene (OSK), which remains intact and is not cleaved by AzoR (Figure 2).

### 3.2. Thermodynamic Analysis of OSK Binding to BLVRB

Isothermal titration calorimetry (ITC) is employed to directly measure the heat generated during BLVRB-OSK complex formation at constant temperature, enabling the characterization of thermodynamic quantities such as ΔH and ΔS. The dissociation constant (K_D_) of OSK is found to be 140 nM (Table 1; Figure 3), indicating that OSK serves as a potent and effective binder to BLVRB. Comparative analysis with previously identified xanthene-based inhibitor, phloxine B, reveals that OSK exhibits tighter binding affinity. Furthermore, OSK exhibits more specific interactions with BLVRB compared with xanthene-based inhibitors, as evidenced by the enthalpic contribution observed in the ITC data (Table 1). The binding of OSK is primarily driven by enthalpic changes, reflecting specific interactions between BLVRB and the compound. The disparity in thermodynamic values between OSA and OSK can be elucidated by the substitution of the diazenyl bond with an alkene bond. In summary, the ITC findings support the presence of specific interactions between BLVRB and OSK.

### 3.3. Investigating the Binding Site of Olsalkene (OSK) in BLVRB Using NMR Spectroscopy

Protein-based NMR spectroscopy stands out as a highly effective method for identifying intermolecular interactions, including those involving small molecules and proteins. Additionally, protein-based NMR methods furnish valuable insights into binding sites, crucial for optimizing the positioning of small molecules within complexes. Typically, this involves monitoring chemical shift changes in ^1^H-^15^N HSQC spectra. Assigning amide resonances is imperative for interpreting HSQC spectra, a task facilitated by HNCA and HN(CO)CA spectra (Appendix A), which provide 95% of backbone resonance assignments. Consequently, we conducted HSQC experiments utilizing ^15^N-labeled BLVRB, titrating OSK to monitor chemical shift changes upon binding. Notably, the FMN binding site in BLVRB was utilized for OSA binding, with Ser111 previously identified as a key residue in BLVRB’s catalytic mechanism. Thus, we monitored the amide chemical shift changes in these residues in HSQC spectra and observed that OSK binding to BLVRB induced shifts in the amide resonances of the FMN binding site (Figure 4A). Additionally, verification of the drug’s binding effect on the BLVRB surface was successfully accomplished by analyzing chemical shift perturbations (CSPs) and mapping them onto the BLVRB surface structure (Figure 4D). This analysis clearly identified the FMN binding site, particularly the xanthene ring of FMN, as the binding site of OSK.

### 3.4. X-ray Crystal Structure of BLVRB-OSK Complex

The crystal structures of BLVRB bound to NADP^+^ and OSK were resolved at a resolution of 1.7 Å (Appendix A). The crystals belong to space group P21212, with each crystal asymmetric unit containing two BLVRB molecules (designated as Mol-A and -B). The heavy atom root mean square deviation (RMSD) between Mol-A and Mol-B varies by 1.91 Å. Notably, a specific region of BLVRB, situated within the vicinity of crystal contacts, exhibits considerable disparity between Mol-A and Mol-B. Particularly, the structures of drug-binding pockets display distinctions between Mol-A and Mol-B. While the OSK-binding pocket structure in Mol-B is influenced by the presence of the second BLVRB molecule due to crystal contacts, the structure of the OSK-binding pocket in Mol-A remains intact in a monomeric state of BLVRB. Furthermore, overlaying the Mol-A complexed with OSK onto Mol-B reveals a spatial collision between its R174 residue and the D130 of the original Mol-A. Additionally, Mol-B, including bound NADP^+^ and drug molecules, has higher crystallographic B factors. Consequently, detailed structural analyses of the complex were focused on Mol-A. 

The BLVRB-OSK complex structure resembles previously determined structures with RMSDs of 1.06 and 0.17 Å compared with the BLVRB-NADP^+^-FMN (PDB ID: 1HE4) and BLVRB-NADP^+^-OSA (PDB ID: 7ERA) complexes, respectively. This indicates a high degree of structural similarity between BLVRB in complex with OSK and BLVRB complexed with its native substrate and OSA. The binding site of OSK closely associates with the nicotinamide moiety of the NADP^+^ cofactor, where the xanthene ring of FMN adopts a similar orientation to xanthene-based inhibitors like phloxine B and erythrosine, as evidenced by NMR titration experiments (Figure 4). A primary driving force for the binding of OSK is the ring-stacking interaction between the nicotinamide moiety of NADP^+^ and OSK, along with hydrophobic interactions involving residues S111, F113, W116, L125, V128, P152, and H153 (Figure 5B). Additionally, favorable interactions are facilitated by positively charged residues R78, K120, R124, R170, and K178. Notably, OSK displaces the sidechain of W116 (as observed in the FMN complex) and shifts it toward the binding pocket by approximately 2.5 Å. These observations align well with the binding affinity measured via ITC (Figure 5 and Table 1).

### 3.5. BLVRB Enzymatic Kinetics with Inhibitors

In examining the enzymatic kinetics of BLVRB in the presence of NADPH with inhibitors OSK and OSA, the Lineweaver–Burk plot reveals a mixed inhibition pattern characterized by competitive (K_ic_) and uncompetitive (K_iu_) inhibition constants (Figure 6). Starting with OSK, the Lineweaver–Burk plot illustrates a maximum velocity (V_m_) of 4.80 ± 0.16 µM/min and a Michaelis constant (K_m_) of 61.73 ± 5.28 µM. The competitive inhibition constant (K_ic_) of 0.56 ± 0.10 µM suggests that OSK competes with the substrate (FMN) for binding to the active site of BLVRB. Meanwhile, the uncompetitive inhibition constant (K_iu_) of 0.61 ± 0.06 µM indicates that OSK binds to both NADP^+^ and NADPH-complexed BLVRB (Figure 6A), altering its conformation and inhibiting enzymatic activity. Moving to OSA, similar kinetics are observed with a V_m_ of 4.71 ± 0.09 µM/min and a K_m_ of 61.57 ± 2.75 µM. The competitive (K_ic_ = 0.55 ± 0.04 µM) and uncompetitive (K_iu_ = 0.79 ± 0.06 µM) inhibition constants suggest that OSA also competes with the substrate for binding to the active site and interacts with both NADP^+^ and NADPH-complexed BLVRB. Overall, the mixed inhibition pattern observed for OSK and OSA suggests that these inhibitors can bind to both the NADP^+^ and NADPH-complexed BLVRB, affecting BLVRB activity. Furthermore, both OSK and OSA are able to bind to the apo-BLVRB distinct from the xanthene-based inhibitor, phloxine B, implying better inhibitory activity of OSK and OSA (Appendix A). Taken together, these results show the kinetic mechanism of BLVRB inhibition with OSK and OSA (Figure 6A). Understanding the detailed kinetics of these inhibitors is essential for unraveling the regulatory mechanisms of BLVRB and may have implications for drug development targeting this enzyme.

## 4. Conclusions

In brief, our study successfully developed a novel inhibitor of BLVRB, substituting the diazenyl bond of OSA with an alkene bond to avoid cleavage by AzoR in vivo. We characterized the binding properties of OSK using a combination of biochemical and biophysical techniques, including NMR spectroscopy, isothermal titration calorimetry, and X-ray crystallography. OSK exhibited high affinity and specific binding to the substrate binding pocket of BLVRB with a K_D_ of 0.140 µM, akin to the previously studied OSA. Through NMR analysis, we confirmed that the binding pocket of OSK closely resembles that of BLVRB’s natural partner, FMN, and involves an important residue for BLVRB enzymatic activity, Ser111.

Unlike certain xanthene-based inhibitors, which may induce BLVRB multimerization, potentially leading to treatment complications, OSK demonstrated selective binding to BLVRB without such adverse effects. Through a comprehensive analysis involving NMR, ITC, and crystallography, we elucidated the mechanism of action of OSK. Specifically, we identified hydrophobic interactions and ring stacking between NADP^+^ and the inhibitor, as well as favorable electrostatic interactions with the sidechain of R78 in BLVRB, contributing to increased binding affinity. 

Moving OSK to clinical trials requires extensive preclinical studies to evaluate its pharmacokinetics, toxicity, and efficacy. Given its specificity and high binding affinity to BLVRB, OSK shows promise as a therapeutic agent, particularly for diseases involving oxidative stress and impaired heme metabolism. The next steps would involve optimizing its formulation, determining appropriate dosing, and conducting safety assessments in animal models before initiating Phase I clinical trials. We also anticipate that this novel inhibitor, OSK, which is resistant to cleavage by AzoR, will provide a valuable platform for further investigations into the physiological functions of BLVRB, such as its role in ROS accumulation and megakaryocyte differentiation. 

## Data Availability

The original contributions presented in the study are included in the article, and further inquiries can be directed to the corresponding authors.

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
