# Peer review of "An Innovative Inhibitor with a New Chemical Moiety Aimed at Biliverdin IXβ Reductase for Thrombocytopenia and Resilient against Cellular Degradation"

_pharmaceutics, 2024, doi:10.3390/pharmaceutics16091148_

Round 1

Reviewer 1 Report

Comments and Suggestions for Authors

This paper by Jung et al aims at devised inhibitor of  biliverdin IXβ reductase based on olsalazine with improved stability against degradation by AzoR. The manuscript is overall decently written and interesting, but I do have some comments to make:

1. I did not see information about the purity and how it was determined of the olsalkene

2. Will be great indicating how the concentration and purity of the expressed proteins were assessed (AzoR and BLVRB)

Minor comments: The list of abbreviations is incomplete

Author Response

Reviewer 1

This paper by Jung et al aims at devised inhibitor of  biliverdin IXβ reductase based on olsalazine with improved stability against degradation by AzoR. The manuscript is overall decently written and interesting, but I do have some comments to make:

  1. I did not see information about the purity and how it was determined of the olsalkene

We have determined the purity (>99%) using NMR and mass spectrometer. We added the purity in the 2. Materials and Methods section, under the 'Chemical synthesis of olsalkene (OSK)' subsection, at line 140.

  1. Will be great indicating how the concentration and purity of the expressed proteins were assessed (AzoR and BLVRB)

BLVRB contains one tryptophan and six tyrosine residues. The concentration was determined using UV absorption with an extinction coefficient (A280) of 14,440 M-1⋅cm-1. The purity was confirmed by SDS-PAGE gel analysis. In the case of AzoR, only partial purification was performed, and the experiment was designed solely to confirm whether olsalazine’s azo group could be cleaved by AzoR, regardless of its purity. We have included this description in the 2. Materials and Methods section, under the ‘Expression and purification of BLVRB' subsection, at line 200/202.

Minor comments: The list of abbreviations is incomplete

We have added more abbreviations.

Reviewer 2 Report

Comments and Suggestions for Authors

In this manuscript, Hoe-Myung Jung et al provide a detailed biophysical characterisation of olsalkene, a modified version of the BLVRB inhibitor olsalazine which is less susceptible to cleavage. Considering the positive effect BLVRB has on platelet generation, this could be an interesting and useful molecule for the treatment of thrombocytopenic patients. The biophysical characterisations performed in this study are critical for understanding how olsalkene functions and is necessary for its future development. However, improvements to the explanation of results, impact of the research and future direction of the work is required. Specific comments are provided below. 

1)      Can the authors comment on how the concentrations of AzoR used in the cleavage assay of olsalazine relate to physiological concentrations in the body?

2)      I believe supplemental methods were not attached correctly as I have not access to them.

3)      No details of methodology for NMR in manuscript. This must be added. It is also unclear if backbone assignments were performed here, or if previously used assignments for the HSQC were used.  Details may be in supplementary which I cannot see but this needs to be referenced in the main text also.

4)      Purification gels of BLVRB should be shown in the supplementary. Additionally, supplementary figure 1 S1 should be more clearly labelled to highlight AzoR on the gel and explain what each purification step represents

5)      Authors should show the 1H15N HSQC of apo BLVRB spectrum in supplementary to show that apo BLVRB is correctly folded.

6)      ITC curves saturate very quickly (figure 3) which suggests the concentration of OSK and OSA are too high. I suggest the authors attempt the same experiment but use 0.5 mM ligand and not 1 mM. This would improve the accuracy of the affinity calculations without too much effort.  

7)      Figure legends are lacking key details that make it hard to follow. In particular the figure 6 legend has no detail and cannot be understood in isolation. What is this figure showing and what is the proposed inhibition mechanism? Furthermore, Figure 3 should detail protein concentrations and affinities calculated in ITC, colours of different molecules should be more accurately described in figure 5 (yellow stick molecule not mentioned).

8)      It would be more accurate and convincing to compare the binding of 5-ASA vs OSK and OSA to BLVRB using ITC as shown in figure 3.

9)      I do not think it necessary to include the reference data for phloxine B and olsalazine in table 1 and instead can just be mentioned in the text. The previous ITC data for olsalazine does not need to be mentioned as it does not correlate with the data generated in this manuscript.

10)  Can the authors explain why half-holo BLVRB was used in the titration studies?

11)      Residues should be labelled onto figure 4 C and D, particularly ser111 and the other major shifted residues in the CSP graph.

12)      Authors should provide a crystallography table with data collection and refinement statistics. Once again, this may be in supplemental methods but I do not have this file.

13)      Can the authors overlay the binding sites of OSK determined by crystallography and NMR to show how closely these two methods predict the binding site.

14)      Can the authors comment on future progression of OSK to the clinic? What are the implications on the half-life of OSK vs OSA in vivo? Also what could the off-target effects be of inhibiting BLVRB outside of promoting platelet generation. The authors should comment on what future work is needed.

15) The following sentence is mis-leading as hemophilia and thrombocytopenia are not the same thing. Please remove mention of hemophilia.

    Introduction paragraph 1. Line 43/44

‘thrombocytopenia, a genetic disorder characterized by increased bleeding (hemophilia) due to low platelet counts’

Comments on the Quality of English Language

English is appropriate for the study. 

Author Response

Reviewer 2

In this manuscript, Hoe-Myung Jung et al provide a detailed biophysical characterisation of olsalkene, a modified version of the BLVRB inhibitor olsalazine which is less susceptible to cleavage. Considering the positive effect BLVRB has on platelet generation, this could be an interesting and useful molecule for the treatment of thrombocytopenic patients. The biophysical characterisations performed in this study are critical for understanding how olsalkene functions and is necessary for its future development. However, improvements to the explanation of results, impact of the research and future direction of the work is required. Specific comments are provided below.

1)      Can the authors comment on how the concentrations of AzoR used in the cleavage assay of olsalazine relate to physiological concentrations in the body?

We have added the relevant content to the Introduction and included the reference.

“It has also been reported that olsalazine has a plasma elimination half-life of approximately 1 hour.”

We have added the sentence in the 1. Introduction section at line 89/90.

2)      I believe supplemental methods were not attached correctly as I have not access to them.

We regret that the supplementary data was not accessible. We have uploaded it again

3)      No details of methodology for NMR in manuscript. This must be added. It is also unclear if backbone assignments were performed here, or if previously used assignments for the HSQC were used.  Details may be in supplementary which I cannot see but this needs to be referenced in the main text also.

We regret the missing information regarding the NMR experiments section, which has now been added at the end of the 2. Materials and Methods section.

4)      Purification gels of BLVRB should be shown in the supplementary. Additionally, supplementary figure 1 S1 should be more clearly labelled to highlight AzoR on the gel and explain what each purification step represents

In Figure S1A, the region presumed to contain AzoR has been marked. The band for Fraction 1 is also indicated because, although NADH reduction was observed in the 100X dilution experiment, the reaction rates for Fractions 3 and 4 were too rapid, resulting in only end-point data. Additionally, Figure S1B shows that the experiment had to be conducted with a 1000X dilution, which makes it look like Fraction 1 lacks AzoR.

5)      Authors should show the 1H15N HSQC of apo BLVRB spectrum in supplementary to show that apo BLVRB is correctly folded.

We apologize for any inconvenience that the reviewer could not access the supplementary information. Figure S3A, B, and C have presented the apo-BLVRB spectrum with and without the drug.

6)      ITC curves saturate very quickly (figure 3) which suggests the concentration of OSK and OSA are too high. I suggest the authors attempt the same experiment but use 0.5 mM ligand and not 1 mM. This would improve the accuracy of the affinity calculations without too much effort. 

To prevent the ITC curve from saturating too quickly, the protein:ligand ratio can be maintained while reducing both the protein and ligand concentrations, resulting in a more gradual ITC curve. However, lowering the concentrations also decreases the thermal signal at each injection point, leading to a reduced signal-to-noise ratio and making integration more challenging.

7)      Figure legends are lacking key details that make it hard to follow. In particular the figure 6 legend has no detail and cannot be understood in isolation. What is this figure showing and what is the proposed inhibition mechanism? Furthermore, Figure 3 should detail protein concentrations and affinities calculated in ITC, colours of different molecules should be more accurately described in figure 5 (yellow stick molecule not mentioned).

Figure 3: We have added the statement 'The cell contained 90 μM protein, while the syringe contained 1,100 μM drug (OSK or OSA)' to the figure legend, and the KD value has also been added to the figure.

Figure 5: We have labeled NADP+ with orange sticks in Figure 5 to accurately describe its identity.

Figure 6: We have included more detail description in the figure caption.

8)      It would be more accurate and convincing to compare the binding of 5-ASA vs OSK and OSA to BLVRB using ITC as shown in figure 3.

It seems that the supplemental materials, where we calculated the affinity for 5-ASA using NMR in Figure S3, were not reviewed. We will re-upload the document to ensure the information is accessible.

9)      I do not think it necessary to include the reference data for phloxine B and olsalazine in table 1 and instead can just be mentioned in the text. The previous ITC data for olsalazine does not need to be mentioned as it does not correlate with the data generated in this manuscript.

Since the ITC data could be influenced by condition dramatically. Thus, we have included the reference data for phloxine B and olsalazine in Table 1 to demonstrate that the thermodynamic parameters from the ITC experiments are consistent with previous results.

10)  Can the authors explain why half-holo BLVRB was used in the titration studies?

The titration study aimed to determine the thermodynamic parameters and affinity of OSK. When conducting experiments with the holo form of BLVRB, it is necessary to account for the influence of endogenous substrates like FMN and FAD, which introduces additional variables. To obtain accurate values, the half-holo form of BLVRB was used instead.

11)  Residues should be labelled onto figure 4 C and D, particularly ser111 and the other major shifted residues in the CSP graph.

For better clarity, both Figures 4C and 4D have been adjusted. Additionally, Ser111 has been clearly labeled to make the figure easier to understand.

12)      Authors should provide a crystallography table with data collection and refinement statistics. Once again, this may be in supplemental methods but I do not have this file.

We apologize for the oversight. We will add the crystallography table with data collection and refinement statistics and upload the revised version.

13)      Can the authors overlay the binding sites of OSK determined by crystallography and NMR to show how closely these two methods predict the binding site.

Figure 4B has been updated to show the residues involved in hydrophobic interactions, as indicated by the LigPlot, to demonstrate the correlation between the NMR and crystallography data.

14)      Can the authors comment on future progression of OSK to the clinic? What are the implications on the half-life of OSK vs OSA in vivo? Also what could the off-target effects be of inhibiting BLVRB outside of promoting platelet generation. The authors should comment on what future work is needed.

We have added “Moving OSK to clinical trials requires extensive preclinical studies to evaluate its pharmacokinetics, toxicity, and efficacy. Given its specificity and high binding affinity to BLVRB, OSK shows promise as a therapeutic agent, particularly for diseases involving oxidative stress and impaired heme metabolism. The next steps would involve optimizing its formulation, determining appropriate dosing, and conducting safety assessments in animal models before initiating Phase I clinical trials.”

15) The following sentence is mis-leading as hemophilia and thrombocytopenia are not the same thing. Please remove mention of hemophilia.

We have changed the sentence in the 1. Introduction section at line 43. 

Reviewer 3 Report

Comments and Suggestions for Authors

Authors performed valuable research on design of biliverdin IXbeta reductase inhibitor. They provide study on resistance towards azoreductase, thermodynamic analysis of binding to target enzyme, binding site investigation and other important issues. The manuscript can be published after some corrections in synthetic part.

-          It would be better to put the structure of olsalazine into introduction part, you mention it several times and provide the structure only in the page 6.

-          Please check reaction conditions in the Scheme. First step: conditions (room t?), duration. Second step: reflux for 4 h, and before it 0 to 60 degC for 2 h. The last step: -78degC to room temperature, interaction with HCl after BBr3 and after KOH, without it you couldn’t get free COOH groups.

-          Lines 145-148 NMR data: in the line 145 1 is superscript, in the line 146 2 in 2H -not superscript. Signals of OH and COOH groups?

-          Page 3, synthesis description. The first sentence (lines 114-115) is more suitable to the chapter Chemicals. And data from line 115 to line 122 should be moved before the synthetic procedure, may be to separate paragraph. Yields in % should be given for first and second steps. The amount of solvent is not given (DMF in the line 124, THF in the line 130, DCM in the line 136). What eluent for column chromatography (line 128)? Potassium carbonate (line 133) solution? What concentration of potassium hydroxide solution (line 140)? Anhydrous K2CO3 (line 124), not dried.

In discussion part the separate chapter about synthesis is desirable. You should give short comments on synthetic steps. What about the last one? Type of organic reaction? Reference to similar approach for stilbene derivatives?

Author Response

Reviewer 3

Authors performed valuable research on design of biliverdin IXbeta reductase inhibitor. They provide study on resistance towards azoreductase, thermodynamic analysis of binding to target enzyme, binding site investigation and other important issues. The manuscript can be published after some corrections in synthetic part.

-          It would be better to put the structure of olsalazine into introduction part, you mention it several times and provide the structure only in the page 6.

Thank you for the suggestion. As the reviewer suggested, the presentation of the structure of olsalzine needs to be as early as possible. Thus, Figure 1 shows the structure of olsalazine.

-          Please check reaction conditions in the Scheme. First step: conditions (room t?), duration. Second step: reflux for 4 h, and before it 0 to 60 degC for 2 h. The last step: -78degC to room temperature, interaction with HCl after BBr3 and after KOH, without it you couldn’t get free COOH groups

Based upon the reviewer’s point of observation, we have given the schemes, additionally, the procedure is as follows. To a solution of dimethyl 5,5'-(ethene-1,2-diyl)(E)-bis(2-methoxybenzoate) in DCM at -78° C was added BBr3 (1M DCM) in RB flask, allowed to stir for 5 h. then quenched in 1M HCl. the solvent was removed and resulting solid was washed with water and brine solution, then extracted in ethyl acetate solvent. Once evaporated, the solid was dissolved in 95% EtOH, and KOH (2.5 equiv. 40 mg) was added. The resulting solution was refluxed for 3 h, after precipitation occurred in the reaction, we allowed to keep under room temperature, acidified with 1M HCl, finally the crude mass extracted and recrystallized with ethyl acetate respectively.

-          Lines 145-148 NMR data: in the line 145 1 is superscript, in the line 146 2 in 2H -not superscript. Signals of OH and COOH groups?

In accordance to the reviewer suggestion, we have made the corrections in the manuscript.

-          Page 3, synthesis description. The first sentence (lines 114-115) is more suitable to the chapter Chemicals. And data from line 115 to line 122 should be moved before the synthetic procedure, may be to separate paragraph. Yields in % should be given for first and second steps. The amount of solvent is not given (DMF in the line 124, THF in the line 130, DCM in the line 136). What eluent for column chromatography (line 128)? Potassium carbonate (line 133) solution? What concentration of potassium hydroxide solution (line 140)? Anhydrous K2CO3 (line 124), not dried.

We have moved the sentence to the chemical section at line 106/107 as per your suggestion. We have also separated the paragraph according to your suggestion. As per the reviewer comments, the sentence to be modified, the yields of the first step is 95% and the second step is 15%; 10 mL of DMF, 10 mL of THF and 10 mL of DCM have been utilized. For column chromatography, we used hexane:EtOAc (1:4), potassium carbonate dissolved in water (saturated) as for quenching has denoted as solution. The KOH (2.5 equiv. 40 mg) has used. We have changed the sentence in the 2. Materials and Methods section at line 121, 125, 131.

- In discussion part the separate chapter about synthesis is desirable. You should give short comments on synthetic steps. What about the last one? Type of organic reaction? Reference to similar approach for stilbene derivatives?

As per the reviewer’s suggestion, we have given the short comments in the supplementary information as stilbene based synthetic compounds have gained the predominant place in pharmaceutical along with material chemistry domain.

For instance, stilbene and its related motifs hold enormous potential applications, due to their diverse spectrum of biological applications such as anticancer, antiproliferative, antiangiogenesis, antimicrobial, antileukemic anti-HIV and similarly for industrial purpose as electrochemical, dyes, dye laser, coloring textiles, organic LED, fluorescent and optical brightners properties etc.

Due to the non-availability of naturally occurring stilbenes in adequate quantities, the development of new synthetic methodologies required for their preparation at large scale, Like Wittig or Horner–Wadsworth–Emmons (HWE) olefination, Perkin aldol condensation, coupling reactions based on Suzuki– Miyaura, Sonogashira, Stille, Mizoroki–Heck, Negishi, Grubbs, McMurry, Knoevenagel–Doebner, Ramberg–Bucklund reactions etc. Despite, many synthetic strategies for the syntheses of stilbene scaffolds and related structures are available in the literatures. Among them, we have used a BBr3 (Lewis acid) has been generally used for the demethylation of methyl aryl ethers.